# Exposure to *Treponema pallidum* Infection among Adolescent and Young Adult Women in Roraima, Amazon Region of Brazil

**DOI:** 10.3390/microorganisms11102382

**Published:** 2023-09-23

**Authors:** Maria Eduarda de Sousa Avelino, Andrio Silva da Silva, Leonardo Gabriel Campelo Pinto de Figueiredo, Ricardo Roberto de Souza Fonseca, Cláudia Ribeiro Menezes, Sandra Souza Lima, Ana Luísa Mendes, Carla Hart Borges da Silva, Isabela Vanessa Sampaio dos Reis, Huendel Batista de Figueiredo Nunes, Bianca Jorge Sequeira, Luiz Fernando Almeida Machado

**Affiliations:** 1Biology of Infectious and Parasitic Agents Post-Graduate Program, Federal University of Pará, Belém 66075-110, PA, Brazil; mdudasc@hotmail.com (M.E.d.S.A.); ricardofonseca285@gmail.com (R.R.d.S.F.); claudiarm@ufpa.br (C.R.M.); 2Virology Laboratory, Institute of Biological Sciences, Federal University of Pará, Belém 66075-110, PA, Brazil; chelseafc200966@hotmail.com (A.S.d.S.); lfigueireddo@gmail.com (L.G.C.P.d.F.); andra.souza.lima@gmail.com (S.S.L.); 3Health Research Center, School of Medicine, Federal University of Roraima, Boa Vista 69317-810, RR, Brazil; luisaanamendes@gmail.com (A.L.M.); carlahartborges@hotmail.com (C.H.B.d.S.); isaa.vsreis@gmail.com (I.V.S.d.R.); huendellbfnunes@gmail.com (H.B.d.F.N.); bianca.costa@ufrr.br (B.J.S.)

**Keywords:** *Treponema pallidum*, prevalence, amazon region, Brazil

## Abstract

Background: Syphilis is a chronic infectious disease, and its prevalence has been described since the 15th century. Because of the high prevalence of this infection in Brazil, this study aimed to evaluate the prevalence of syphilis and its associated factors among adolescent and young women living in the city of Boa Vista, Roraima, Brazil. Methods: The present study was cross-sectional, descriptive, analytical and quantitative. It involved 200 young and adolescent women. Laboratory tests were performed to diagnose syphilis, and a sociodemographic and epidemiological questionnaire was employed. Results: In the studied sample, 10 women had a positive result for syphilis, characterizing a prevalence of 5% for infection with *Treponema pallidum*. There was a statistically significant association between a monthly family income of less than 1 minimum wage and syphilis (*p* = 0.0449) and between illicit drug use and syphilis (*p* = 0.0234). Conclusions: These results indicate the need for public health interventions, action plans, and the implementation of risk reduction strategies focused on this population.

## 1. Introduction

Syphilis is a chronic infectious disease, and its prevalence has been described since the 15th century [1]. It is caused by the bacteria *Treponema pallidum* (*T. pallidum*) [1,2]. Syphilis is transmitted through direct contact with infected body fluids from mucosal lesions during sex or via vertical transmission in pregnancy (congenital syphilis) [3]. With infectivity up to 10–30% for sexual contact or 60% among people in relationships, the main clinical manifestations result from local inflammatory responses of replicating spirochetes [4,5,6].

The epidemiology of syphilis varies worldwide. Differing patterns are noted. Syphilis is a global public health problem [7,8]. Syphilis occurs mainly in countries with limited resources associated with low investments in health primary care. Syphilis is among the most common sexually transmitted infections (STIs), mostly transmitted by unprotected sexual practices and in pregnancy from the mother to the foetus, and can progress into more severe stages in the absence of treatment, from primary syphilis to secondary or tertiary; syphilis also presents a latency period [9].

According to the Brazilian Ministry of Health, an increase in registered syphilis cases was observed over the last ten years in its 2018 annual report. In 2017, 119,800 cases of acquired syphilis were reported, a rate of 58.1 cases per 100,000 inhabitants, which is considered very high [10]. Brazil is divided into five regions based on geographical features (North, Northeast, Southeast, South and Central-West) and 5570 municipalities. The distribution of syphilis among these regions and municipalities is highly heterogeneous, and municipalities with a larger population usually have higher probabilities of presenting an epidemic profile for syphilis compared to other municipalities [11]. Roraima is a state in the Brazilian Amazon that borders Venezuela and the Cooperative Republic of Guyana and is known as the last Brazilian frontier, located in the extreme north of the country. Boa Vista is the capital of the state of Roraima and has 265,000 inhabitants, a concentrated population representing 65% of all inhabitants of the state [12].

The prevalence of syphilis in a young population can be a predictor of important failures in health services. Therefore, it is essential to analyse the epidemiological profile of the disease to guide new public health policies and improve current policies, enabling the monitoring of health indicators for strategic planning, especially with regard to the evaluation of interventions aimed at reducing these indicators. Thus, this study aimed to evaluate exposure to *Treponema pallidum* among adolescents and young adults in Roraima, Amazon region of Brazil.

## 2. Materials and Methods

### 2.1. Type of Study and Ethical Aspects

The present study was cross-sectional, descriptive, analytical and quantitative. It involved 200 young and adolescent women residing in Boa Vista, Roraima. Laboratory tests were performed to diagnose syphilis, and a sociodemographic and epidemiological questionnaire was employed. The Research Ethics Committee with Human Beings of the Institute of Health Sciences of the Federal University of Pará, Brazil approved this study under protocol number 3,297,951.

### 2.2. Study Design and Sample Size

The study group included 200 women who attended a basic health unit in Boa Vista between July 2019 and December 2020. The recruitment of participants was carried out by spontaneous solicitation during cervical cancer screening consultations. All participants went to the basic health unit to undergo a preventive exam for cervical cancer. During the consultation, they were invited to participate in this study, and those who accepted signed a consent form. All participants were duly informed about this study before signing the consent form. Then, they answered a sociodemographic and epidemiological semistructured questionnaire and underwent laboratory tests to diagnose syphilis.

The inclusion criteria were as follows: aged between 14 and 29 years, female, Brazilian, and agreed to participate in this study by signing the informed consent form and answering the questionnaire. The exclusion criteria were indigenous individuals and people with cognitive impairment who were unable to answer the questionnaire.

The subjects who agreed to participate in this research signed the consent form, and information such as age, sex, marital status, schooling, monthly income, condom use in sexual intercourse, sexual orientation, use of illicit drugs and alcoholic beverages and history of STIs were obtained through a semistructured questionnaire.

The determination of the sample size was based on sampling calculations using BioEstat 5.0 software, using sample size calculations for proportions with a test power of 0.90 and an alpha level of 0.05. Thus, the total sample size was 197 individuals.

### 2.3. Laboratory Tests and Statistical Analysis

To evaluate the prevalence of syphilis among the study participants, a rapid immunochromatographic test was used as an initial screening method for the detection of anti-treponemal antibodies (Imuno-rapido Sífilis; Wama Diagnóstica) for the qualitative determination of antibodies (IgG and IgM) against *Treponema pallidum* [13].

In the samples that showed positive results, the Venereal Disease Research Laboratory (VDRL) (Wama Diagnóstica) performed a nontreponemal flocculation test to verify active infection or serological scarring using pure and diluted serum/plasma at a ratio of 1:8 [14]. For the reactive samples in the qualitative test, quantitative analysis was performed, and the biological sample was diluted in proportions of 1:4, 1:8, 1:16, and 1:32 until the dilution verified positivity. For each dilution, the sample was placed in a different well of the plate, and the test was carried out as recommended by the manufacturer. The result of the reaction was given by the highest dilution in which the test was still reactive.

The rapid tests were performed at the basic health unit, and the participants who tested positive for syphilis had 5 ml of their venous blood collected. The venous blood sample was transported to the bioassay laboratory of the Postgraduate Program in Natural Resources at the Federal University of Roraima to perform VDRL testing.

All statistical procedures were performed in SPSS 21.0 for Windows (SPSS Inc., Amonk, NY, USA). Descriptive analysis of the data was performed, with distribution of relative frequencies, and then the data were categorized and grouped. Then, Fisher’s exact test and the G test were performed, with a significance level of 5% (*p* < 0.05).

## 3. Results

When analysing the sociodemographic characteristics of the study participants, the majority were over 22 years old (67.5%), were married (47.5%), had more than 8 years of schooling (92.0%) and had a monthly family income greater than 1 minimum wage (56.0%).

Regarding behavioural characteristics, 84.5% of participants were heterosexual, 81.0% had never used illicit drugs, 52.0% consumed alcoholic beverages, 66.5% had no tattoos, 68.5% did not have anal sex, 78.5% had only one sexual partner at the time of this study, 78.5% had no history of sexually transmitted infections, and only 24.5% reported using condoms at all times (Table 1).

Among the 200 women evaluated in this study, 10 had a positive result for syphilis, characterizing a prevalence of 5% of *Treponema pallidum* infection. When evaluating the association between the sociodemographic variables and the positive results for syphilis, a statistically significant association was evidenced only between having a family income less than 1 minimum wage and having syphilis (*p* = 0.0449). There was no association between age group, marital status or level of education and syphilis. Regarding VDRL titres, among our samples, the median titre was a 1:8 dilution, although no association was significant among our variations and VDRL titres.

On the other hand, among the behavioural variables, there was only a statistically significant association between illicit drug use and syphilis (*p* = 0.0234). The use of alcoholic beverages, having tattoos, using or not using condoms during sexual intercourse, practising anal sex, having multiple sexual partners or having previously had an STI were not associated with the final outcome.

## 4. Discussion

To the best of the authors’ knowledge, this is the first study in the literature to evaluate a younger population in North Brazil with syphilis, and our results, even with a small sample, evidenced a higher prevalence of *Treponema pallidum* infection in a poor population. The epidemiology of syphilis varies worldwide. Differing patterns are noted, and syphilis remains a global public health problem. In high-income and middle-income countries, the majority of cases occur among men who have sex with men (MSM) [15,16]. However, in the United States, from 2013 to 2017, the incidence of primary and secondary syphilis increased by 72.2%. In 2017 alone, it increased by 10.5% (9% among men and 21.2% among women). The increase among women was particularly problematic because it was accompanied by a concomitant increase in congenital syphilis [17,18]. Therefore, the present study aimed to study behavioural and sociodemographic factors associated with syphilis in women because the prevalence of syphilis continues to be a major public health problem.

In this study, the prevalence of syphilis found among women was 5%. This result is higher than that found in the general population of Roraima (3.2%) and almost equal to the national prevalence (5.2%) [19,20]. Another study conducted in Paraná also found a lower prevalence (4.3%) [21].

A multicentre study conducted in 10 Brazilian cities involving men and women found a prevalence of 13.9%, with higher rates in Rio de Janeiro (23.5%) and Belo Horizonte (13.9%) [22]. A study conducted in the São Paulo (Brazil) homeless population, which evaluated the prevalence of syphilis and associated factors among 1405 individuals, revealed a prevalence of 7% [23]. Another study carried out in Angola, Africa, among individuals attending a centre of reference and testing for HIV showed higher positivity for syphilis (15%) among the 431 research subjects [24]. Marchese et al. [25] also demonstrated a higher prevalence of syphilis, the same rates as those in our study, although in the authors’ study, the population was different. In our study, young females were analysed, and Marchese et al. evaluated the prevalence of HIV-syphilis coinfections mostly among men who have sex with men.

In the present study, no association was found between age group and syphilis. However, all participants were adolescent or young women aged 14–29 years. This is because young people also showed a greater increase in the incidence of syphilis; among women, the 15–24-year-old group had the highest rates. In the United States, it was reported that 45% of incident STI cases are concentrated in the population aged 15–24 years [26,27]. On the other hand, other studies found a higher prevalence of syphilis among women over 30 years of age [28,29,30].

There was also no association between a positive diagnosis for syphilis and education level or marital status. This result differs from those of other studies that indicated that syphilis is associated with a low education level and single, divorced or separated marital status [21,28,29,30]. Therefore, according to most of the literature, not having a steady sexual partner is a risk factor for syphilis, a fact not observed in this study.

In the present study, an association between low monthly family income and syphilis was observed. There was an association between earning a minimum wage and having syphilis. The minimum wage in Brazil is currently R$1230.00, which is equivalent to US$250. This result is corroborated by other authors who point to women’s place in society as a strong predictor for infection with *Treponema pallidum*, arguing that the prevalence of syphilis decreases according to the increase in family income [29,31,32,33,34]. This association is worrying because Brazil is a country where the majority of the population has low income, and the city of Boa Vista-Roraima is located in the northern region, one of the poorest regions of the country.

Lifestyle and behavioural factors (number of sexual partners, partner characteristics, use of condoms, sexual practices and demand for medical care) are variables that possibly contribute to the wide variation in the prevalence of syphilis [33,34]. However, there was only a statistically significant association between using illicit drugs and having syphilis.

Although the use of condoms in all sexual encounters was reported by few women (24.5%), no association was found between nonuse and a positive result for syphilis in this study or between a history of STIs and having syphilis. This finding differs from those in several other studies that indicated that these two variables are risk factors for syphilis [28,31,35]. Additionally, our treponemal serologic test was demonstrated to be accurate for the diagnosis of syphilis, especially in cases with a titre of 1:8; therefore, in our Brazilian Amazon region, this test should be used to increase testing results and improve treatment in these positive young populations.

## 5. Conclusions

This study detected a 5% prevalence of *Treponema pallidum* infection among adolescent and young women in the city of Boa Vista, Roraima, Amazon region of Brazil. Associations were observed between the use of illicit drugs and a monthly family income of less than 1 minimum wage and having syphilis. These results indicate the need for public health interventions, action plans, and the implementation of risk reduction strategies focused on this population. This study is expected to stimulate public policies for health promotion and prevention of syphilis in the female population of Roraima.

## Figures and Tables

**Table 1 microorganisms-11-02382-t001:** Exploratory analysis of potential factors associated with exposure to *Treponema pallidum* among adolescents and young people in Roraima State, northern Brazil.

	Total (*n* = 200)		Exposure (*n* = 10)		No Exposure (*n* = 190)		*p*-Value
Characteristics	*n*	%	*n*	%	*n*	%	
Age (years)							
<15	3	1.5	0	0.0	3	100	0.5545 ^b^
16–22	62	31.0	5	8.1	57	91.9	
>22	135	67.5	5	3.7	130	96.3	
Marital status							
Single	94	47.0	7	7.4	87	92.6	0.2291 ^b^
Married	95	47.5	2	2.1	93	97.9	
Divorced/Widowed	11	5.5	1	9.1	10	90.9	
Schooling (years of study)							
Up to 8 years	16	8.0	-	-	16	100.0	0.6108 ^a^
>8 years	184	92.0	10	5.4	174	94.6	
Monthly income *							
Up to one Brazilian minimum wage	88	44.0	1	1.1	87	98.9	0.0449 ^a^
More than one Brazilian minimum wage	112	56.0	9	8.0	103	92.0	
Sexual orientation							
Homosexual/bisexual/trans	31	15.5	-	-	31	100.0	0.2286 ^a^
Heterosexual	169	84.5	10	5.9	159	94.1	
Use of illicit drugs							
Yes	38	19.0	5	13.2	33	86.8	0.0234 ^a^
No	162	81.0	5	3.1	157	96.9	
Use of alcohol							
No	96	48.0	2	2.1	94	97.9	0.1360 ^b^
Sometimes	54	27.0	3	5.6	51	94.4	
Yes	50	25.0	5	10.0	45	90.0	
Tattoo							
Yes	67	33.5	3	4.5	64	95.5	1.000 ^a^
No	133	66.5	7	5.3	126	94.7	
Condom use during sexual intercourse **							
No	59	29.5	2	3.4	57	96.6	0.2895 ^b^
Sometimes	92	46.0	7	7.6	85	92.4	
Yes	49	24.5	1	2.0	48	98.0	
Anal sex **							
Yes	63	31.5	4	6.4	59	93.6	0.7278 ^a^
No	137	68.5	6	4.4	131	95.6	
Number of partners per week **							
1	157	78.5	7	4.5	150	95.5	0.3242 ^b^
1–4	30	15.0	3	10.0	27	90.0	
>4	13	6.5	-	-	13	100.0	
STI History ^##^							
Yes	43	21.5	2	4.7	41	95.3	1.000 ^a^
No	157	78.5	8	5.1	149	94.9	

^a^ Fisher’s exact test; ^b^ G test; and *Average of the Brazilian minimum wage in 2019 = BRL 998.00 (equivalent to USD205.83). ** In the last 30 days.^##^ STI: Sexually transmitted infection.

## Data Availability

All data referred to this study is available on the manuscript.

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
