# Peer review of "Exposure to Treponema pallidum Infection among Adolescent and Young Adult Women in Roraima, Amazon Region of Brazil"

_microorganisms, 2023, doi:10.3390/microorganisms11102382_

Round 1
Reviewer 1 Report
You address an important topic, the spread of syphilis among women in Brazil. It is easy to read and understand although some polishing of the English is needed.
I have a serious concern about your sampling procedure and estimation. For example, you state in line 98 that you expected 30% of the sample to be positive for syphilis (or 60 women), but on line 137 you report that only 10 were positive. As a result your power calculations need to be explained.
More important, you state that recruitment was done by "spontaneous demand", which I take to mean that a woman entered for an exam and your agent (nurse/doctor/clerk?) included or not the patient from the study. That is very not rigorous and if the problem is in the manner in which you reported the protocol, then you should be much more specific. If it was as casual as the text makes it appear, you need to rethink and perhaps redo the study.
As well, you do not mention when these samples were taken and over what time period. To what extent did the sampling period overlap the pandemic? Could that have affected who came for gynecological exams during the sampling period?
On a lesser scale of gravity, you chose not to include indigenous women. Why not? They suffer from syphilis as well and were subject over the Bolsonaro years to systematic abuse from poachers and other gangsters. As well, you do not mention Venezuelan women, who became a measurable fraction of the Roraima population emigrating from that country.
A quibble: on line 111 you refer to an "excavation" on a plate when you meant a "well".
In the discussion, you emphasize studies from the United States that treat both men and women. What does that have to do with your study based in a relatively isolated Northern region state. If these comparisons with the US are relevant, you need to clearly show how.
Your finding of the relationship between family income at the level of 1 minimum wage (R$1230) to increased propensity for syphilis infection will have greater impact if you provide a more complete explanation of how low in international terms 1 minimum wage is. Right now, this important finding lacks context and will only be meaningful to clinicians and academics who know Brazil's socio-economic structure.
Needs some work on vocabulary but basically ok
Author Response
I have a serious concern about your sampling procedure and estimation. For example, you state in line 98 that you expected 30% of the sample to be positive for syphilis (or 60 women), but on line 137 you report that only 10 were positive. As a result your power calculations need to be explained.
Question: More important, you state that recruitment was done by "spontaneous demand", which I take to mean that a woman entered for an exam and your agent (nurse/doctor/clerk?) included or not the patient from the study. That is very not rigorous and if the problem is in the manner in which you reported the protocol, then you should be much more specific. If it was as casual as the text makes it appear, you need to rethink and perhaps redo the study.
Answer: The recruitment of participants was carried out by spontaneous demand during a consultation to carry out the preventive examination for cervical cancer. All participants went to the Basic Health Unit to take the preventive exam for cervical cancer. During the consultation, they were invited to participate in the study and those who accepted signed the consent form. All participants were duly informed about the study before signing the consent form. Then they answered a sociodemographic and epidemiological semi-structured questionnaire and were submitted to laboratory tests for the diagnosis of syphilis (lines 86-92)
Question: As well, you do not mention when these samples were taken and over what time period. To what extent did the sampling period overlap the pandemic? Could that have affected who came for gynecological exams during the sampling period?
Answer: The data collection period was between July 2019 and January 2020, therefore no collection was carried out during the COVID-19 pandemic (lines 84-86).
Question: On a lesser scale of gravity, you chose not to include indigenous women. Why not? They suffer from syphilis as well and were subject over the Bolsonaro years to systematic abuse from poachers and other gangsters. As well, you do not mention Venezuelan women, who became a measurable fraction of the Roraima population emigrating from that country.
Answer: We decided not to include indigenous people in the study due to the delay in the processing of projects involving indigenous people in CONEP, since by involving indigenous people, the project would not only be evaluated by a local research ethics committee. Venezuelan women were not included because the Basic Health Unit where the study was conducted is far from the collective shelters where Venezuelans reside. Generally, they are treated in the basic units in the neighborhoods where the shelters are located or at the Armed Forces medical posts located inside the shelters themselves.
A quibble: on line 111 you refer to an "excavation" on a plate when you meant a "well".
Answer: the word excavation has been replaced by well (line 117).
Question: In the discussion, you emphasize studies from the United States that treat both men and women. What does that have to do with your study based in a relatively isolated Northern region state. If these comparisons with the US are relevant, you need to clearly show how.
Answer: North American studies were used in the discussion for two reasons: the existence of few studies on the epidemiological profile of syphilis in states in the north of Brazil and because the studies cited in our discussion evaluated a population sample with characteristics similar to those of the population we evaluated in this study.
Question: Your finding of the relationship between family income at the level of 1 minimum wage (R$1230) to increased propensity for syphilis infection will have greater impact if you provide a more complete explanation of how low in international terms 1 minimum wage is. Right now, this important finding lacks context and will only be meaningful to clinicians and academics who know Brazil's socio-economic structure.
Answer: This information was included in the discussion (lines 190-192)
Reviewer 2 Report
Thank you for submitting this manuscript for consideration. This is a relevant study and is of interest to those working within sexual health. There are some challenges with the presentation and these are outlined as follows:
p1, line 44 - you do not need "Syphilis is a great public health problem worldwide," because you have already stated that it is a "global public health problem" in the sentence before.
p2, Line 47 - is Newman a reference?
p2, line 48 - infection should be infections.
p2, line 48/49 - there is repetition.
p2, line 50 - does not need 'the' before primary syphilis.
p2, line 54 - 100,000
p2, line 59 - municipalities
p2, line 81 - does not need 'capital of the state of Roraima' as this is repetition
p2, line 89 - why were indigenous individuals excluded from taking part?
p2, line 94 - remove 'provided'
p3, very good explanation of power calculation.
p3, line 106 - VDRL should be in parentheses
p3, line 110 - I think it should read "...until the dilution verified positivity."
p3, line 131 - does not need 'sexual relations' at the end and you refer to table 1 but it is labelled as Table 3 on line 132
p4, line 140 - some repetition - I would suggest removing "...using the G test and Fisher's exact test, considering a p value equal to or less than 0.05,".
p5, line 165 - remove Brazil
p5, line 167/169 - I am not sure of the relevance of the Angolan study in relation to your own study.
p5, line 172/173 - reference required to support data from Brazil
p5, line 181 - should read " a fact"
p5, line 191/192 - remove "about lifestyle and behavioural factors"
p5, line 193 - why do you think there was a statistical significance between illicit drug use and having syphilis when the data tells us that there is no significance between unprotected sex and syphilis
I appreciate that this may seem like a lot of comments, but I think they can be addressed easily.
p5, line 173/176 - I do not think this adds anything to your study in Brazil.
Please see the comments above as I have addressed this.
Author Response
Thank you for submitting this manuscript for consideration. This is a relevant study and is of interest to those working within sexual health. There are some challenges with the presentation and these are outlined as follows:
p1, line 44 - you do not need "Syphilis is a great public health problem worldwide," because you have already stated that it is a "global public health problem" in the sentence before.
Answer: This sentence has been replaced in the text (line 46)
p2, Line 47 - is Newman a reference?
Answer: It was a typing error. It has already been removed from de text.
p2, line 48 - infection should be infections.
Answer: This word has been replaced in the text (line 49)
p2, line 48/49 - there is repetition.
Answer: I didn't find it in the text.
p2, line 50 - does not need 'the' before primary syphilis.
Answer: This word has been replaced in the text (line 52)
p2, line 54 - 100,000
Answer: It was replaced in the text (line 55).
p2, line 59 – municipalities
Answer: It was replaced in the text (line 60)
p2, line 81 - does not need 'capital of the state of Roraima' as this is repetition
Answer: This information has been deleted from the text (line 84)
p2, line 89 - why were indigenous individuals excluded from taking part?
Answer: We decided not to include indigenous people in the study due to the delay in the processing of projects involving indigenous people in CONEP, since by involving indigenous people, the project would not only be evaluated by a local research ethics committee.
p2, line 94 - remove 'provided'
Answer: Was removed (line 99).
p3, very good explanation of power calculation.
p3, line 106 - VDRL should be in parentheses
Answer: Got corrected
p3, line 110 - I think it should read "...until the dilution verified positivity."
Answer: It was replaced in the text (line 116)
p3, line 131 - does not need 'sexual relations' at the end and you refer to table 1 but it is labelled as Table 3 on line 132
Answer: It was replaced in the text (lines 137-139).
p4, line 140 - some repetition - I would suggest removing "...using the G test and Fisher's exact test, considering a p value equal to or less than 0.05,".
Answer: ok
p5, line 165 - remove Brazil
Answer: ok
p5, line 167/169 - I am not sure of the relevance of the Angolan study in relation to your own study.
Answer: Population economic characteristics are similar and low income is a factor associated with infection.
p5, line 172/173 - reference required to support data from Brazil
p5, line 181 - should read " a fact"
Answer: ok (line 188)
p5, line 191/192 - remove "about lifestyle and behavioural factors"
Answer: ok (line 200)
p5, line 193 - why do you think there was a statistical significance between illicit drug use and having syphilis when the data tells us that there is no significance between unprotected sex and syphilis
Answer: Unfortunately, the data collected does not allow us to make this inference. It is a strange result, because, in fact, it is common for drug use to be associated with not using condoms.
p5, line 173/176 - I do not think this adds anything to your study in Brazil.
Reviewer 3 Report
Title of the article:
Exposure to Treponema pallidum infection among adolescent and young adult women in Roraima, Amazon Region of Brazil
Manuscript ID:
microorganisms-2518593
General Comments
Thank you for the opportunity to review your interesting manuscript. I enjoyed reading it, although I think that several aspects of the text should be reorganized formally and in the contents (see Major Compulsory Revisions). The most relevant concern that I have is that English syntax is not optimal, and this manuscript would benefit from assistance from a native English speaker (see Major Compulsory Revisions). Moreover, the authors state that the quantitative VDRL test has been performed, however no titers are shown throughout the manuscript (see Major Compulsory Revisions). Lastly, the discussion should be expanded in several areas (see Major Compulsory Revisions). All things considered, I believe that this manuscript should be suitable for publication only after Major and Extensive Revisions.
- Major Compulsory Revisions
Please, try to formally re-organize the manuscript by maintain the correct identity of each chapter (Introduction, Materials and Methods, Results, and Conclusions) and by avoiding repetition. In line 80-81, and 94-95, some results are presented in the Materials and Methods chapter. Moreover, try to re-organize the first Introduction paragraphs (line 34-51) by avoiding repetition (the syphilis transmission route is repeated 3 times, please try to explain thoroughly only once), and. Most of all, by updating the references [Tiecco G, et al. Pathogens. 2021, doi: 10.3390/pathogens10111364]. Please, avoid referring to syphilis as a “simple disease”!
The authors only mention the use of a quantitative test (VDRL) in the Materials and Methods. However, using the “reverse approach” in syphilis diagnosis has several clinical and crucial implications. Please, try to report VDRL titers in the Results, and provide a thorough explanation in the Discussion. Try to highlight eventual previous syphilis treatment, and HIV co-infection. Moreover, compare your results to the most recent available literature [Marchese V, J Clin Med. 2022, doi: 10.3390/jcm11247499].
Please, expand the discussion in order to enlighten the reasons why your results might differ from other analysis reported in literature (for instance: line 163-169, line 177-181).
Quality of written English
English syntax is not optimal. This manuscript would benefit from assistance from a native English speaker.
Author Response
Reply to reviewer #3
1. Concern of the reviewer • General Comments: Thank you for the opportunity to review your interesting manuscript. I enjoyed reading it, although I think that several aspects of the text should be reorganized formally and in the contents (see Major Compulsory Revisions). The most relevant concern that I have is that English syntax is not optimal, and this manuscript would benefit from assistance from a native English speaker (see Major Compulsory Revisions). Moreover, the authors state that the quantitative VDRL test has been performed, however no titers are shown throughout the manuscript (see Major Compulsory Revisions). Lastly, the discussion should be expanded in several areas (see Major Compulsory Revisions). All things considered, I believe that this manuscript should be suitable for publication only after Major and Extensive Revisions.Major Compulsory Revisions: Please, try to formally re-organize the manuscript by maintain the correct identity of each chapter (Introduction, Materials and Methods, Results, and Conclusions) and by avoiding repetition. In line 80-81, and 94-95, some results are presented in the Materials and Methods chapter. Moreover, try to re-organize the first Introduction paragraphs (line 34-51) by avoiding repetition (the syphilis transmission route is repeated 3 times, please try to explain thoroughly only once), and. Most of all, by updating the references [Tiecco G, et al. Pathogens. 2021, doi: 10.3390/pathogens10111364]. Please, avoid referring to syphilis as a “simple disease”!The authors only mention the use of a quantitative test (VDRL) in the Materials and Methods. However, using the “reverse approach” in syphilis diagnosis has several clinical and crucial implications. Please, try to report VDRL titers in the Results, and provide a thorough explanation in the Discussion. Try to highlight eventual previous syphilis treatment, and HIV co-infection. Moreover, compare your results to the most recent available literature [Marchese V, J Clin Med. 2022, doi: 10.3390/jcm11247499]. Please, expand the discussion in order to enlighten the reasons why your results might differ from other analysis reported in literature (for instance: line 163-169, line 177-181). Our response: Dear Reviewer #3, we appreciate your suggestion and concerning, all your suggestions were added to our paper and were highlighted in green for your convenience, please feel free to evaluate, we hope we could get to satisfy you.
Round 2
Reviewer 1 Report
Overall, I appreciate your acceptance of my comments and questions in the first round.
One last issue: my first comment on the power analysis you did with BioEstat in lines 96 - 99 still does not address the inconsistency in prevalence rates. You show 30% here, but in line 141 you state that the prevalence in your sample was 5%. This happens to all of us. Power calculations are always theoretical. But the inconsistency should still be addressed.
Otherwise, good work. My colleagues here in São Paulo will read it with interest.
Author Response
Dear reviewer 1
We would like to thanks for the opportunity to improve our paper, regarding the last issue it was removed from methods section. we hope to fulfill yours expectation.
Best Regards

Reviewer 3 Report
I find the manuscript improved after the revisions.
However, I do suggest an extensive editing of English as English syntax is not optimal, and this manuscript would benefit from assistance from a native English speaker.
English syntax is not optimal, and this manuscript would benefit from assistance from a native English speaker.
Author Response
Dear reviewer 3
We would like to thanks for the opportunity to improve our paper, regarding English syntax it was revised.
